# Hamiltonian descent for composite objectives

**Brendan O'Donoghue**
DeepMind
bodonoghue@google.com

**Chris J. Maddison**
DeepMind / University of Oxford
cmaddis@google.com

## Abstract

In optimization the duality gap between the primal and the dual problems is a measure of the suboptimality of any primal-dual point. In classical mechanics the equations of motion of a system can be derived from the Hamiltonian function, which is a quantity that describes the total energy of the system. In this paper we consider a convex optimization problem consisting of the sum of two convex functions, sometimes referred to as a composite objective, and we identify the duality gap to be the 'energy' of the system. In the Hamiltonian formalism the energy is conserved, so we add a contractive term to the standard equations of motion so that this energy decreases linearly (*i.e.*, geometrically) with time. This yields a continuous-time ordinary differential equation (ODE) in the primal and dual variables which converges to zero duality gap, *i.e.*, optimality. This ODE has several useful properties: it induces a natural operator splitting; at convergence it yields both the primal and dual solutions; and it is invariant to affine transformation despite only using first order information. We provide several discretizations of this ODE, some of which are new algorithms and others correspond to known techniques, such as the alternating direction method of multipliers (ADMM). We conclude with some numerical examples that show the promise of our approach. We give an example where our technique can solve a convex quadratic minimization problem orders of magnitude faster than several commonly-used gradient methods, including conjugate gradient, when the conditioning of the problem is poor. Our framework provides new insights into previously known algorithms in the literature as well as providing a technique to generate new primal-dual algorithms.

## 1 Introduction and prior work

In physics the Hamiltonian function represents the total energy of a system in some set of coordinates (loosely speaking). In the most typical case the coordinates are the position $x \in \mathbb{R}^n$ and momentum $p \in \mathbb{R}^n$, and the Hamiltonian is the sum of the potential energy, a function of the position, and the kinetic energy, a function of the momentum. The equations of motion for the system can be derived from the Hamiltonian. Let us denote the Hamiltonian as $\mathcal{H} : \mathbb{R}^n \times \mathbb{R}^n \to \mathbb{R}$, which we assume is differentiable, then the equations of motion [1] are given by

$$\dot{x}_t = \nabla_p \mathcal{H}(x_t, p_t), \quad \dot{p}_t = -\nabla_x \mathcal{H}(x_t, p_t),$$

where we use the notation $\dot{x}_t := dx_t/dt$. For ease of notation we shall sometimes use $z := (x, p) \in \mathbb{R}^{2n}$ to denote the concatenation of the position and momentum into a single quantity, in which case we can write the Hamiltonian flow as

$$\dot{z}_t = J\nabla\mathcal{H}(z_t), \quad J = \begin{bmatrix} 0 & I \\ -I & 0 \end{bmatrix}, \tag{1}$$

and note that $J^T J = I$ and that $J$ is skew symmetric, that is $J = -J^T$, and so $v^T J v = 0$ for any $v$. It is easy to show that these equations of motion conserve the Hamiltonian since $\dot{\mathcal{H}}(z_t) =$

$\nabla_z \mathcal{H}(z_t)^T \dot{z}_t = \nabla \mathcal{H}(z_t)^T J \nabla \mathcal{H}(z_t) = 0$. This conservation property is required for anything that models the energy of a system in the physical universe, but not directly useful in optimization where the goal is convergence to an optimum. By adding a contractive term to the Hamiltonian flow we derive an ordinary differential equation (ODE) whose solutions converge to a minimum of the Hamiltonian. We call the resulting flow "Hamiltonian descent".

In optimization there has been a lot of recent interest in continuous-time ordinary differential equations (ODEs) that when discretized yield known or interesting novel algorithms [2, 3, 4]. In particular Su *et al.*[5] derived a simple ODE that corresponds to Nesterov's accelerated gradient scheme [6], see also [7]. That work was extended in [8] where the authors derived a "Bregman Lagrangian" framework that generates a family of continuous-time ODEs corresponding to several discrete-time algorithms, including Nesterov's accelerated gradient. This was extended in [9] to derive a novel acceleration algorithm. In [10] the authors used Lyapunov functions to analyze the convergence properties of continuous and discrete-time systems. There is a natural Hamiltonian perspective on the Bregman Lagrangian, which was exploited in [11] to derive optimization methods from symplectic integrators.

In a similar vein, the authors of [12] used a conformal Hamiltonian system to expand the class of functions for which linear convergence of first-order methods can be obtained by encoding information about the convex conjugate into a kinetic energy. Follow-up work analyzed the properties of conformal symplectic integrators for these conformal Hamiltonian systems [13].

Hamiltonian mechanics have previously been applied to several areas outside of classical mechanics [14], most notably in Hamiltonian Monte Carlo (HMC), where the goal is to sample from a target distribution and Hamiltonian mechanics are used to propose moves in a Metropolis-Hastings algorithm; see [15] for a good survey. More recently Hamiltonian mechanics has been discussed in the context of game theory [16], where a symplectic gradient algorithm was developed that converges to stable fixed points of general games.

## 1.1 The convex conjugate

The Hamiltonian as used in physics is derived by taking the Legendre transform (or convex conjugate) of one of the terms in the Lagrangian describing the system, which for a function $f : \mathbb{R}^n \to \mathbb{R}$ is defined as

$$f^*(p) = \sup_x (x^T p - f(x)).$$

The function $f^*$ is always convex, even if $f$ is not. When $f$ is closed, proper, and convex, then $(f^*)^* = f$, and $(\partial f)^{-1} = \partial f^*$, where $\partial f$ denotes the subdifferential of $f$, which for differentiable functions is just the gradient, *i.e.*, $\partial f = \nabla f$ (or more precisely $\partial f = \{\nabla f\}$) [17].

## 2 Hamiltonian descent

A modification to the Hamiltonian flow equation (1) yields an ordinary differential equation whose solutions decrease the Hamiltonian linearly:

$$\dot{z}_t = J \nabla \mathcal{H}(z_t) + z_\star - z_t, \tag{2}$$

where $z_\star \in \operatorname{argmin}_z \mathcal{H}(z)$. This departs from the standard Hamiltonian flow equations by the addition of the term involving the difference between $z_\star$ and $z_t$. One can view the Hamiltonian descent equation as a flow in a field consisting of the sum of a standard Hamiltonian field and the negative gradient field of function $(1/2)\|z_t - z_\star\|_2^2$. Solutions to this differential equation descend the level sets of the Hamiltonian and so we refer to (2) as *Hamiltonian descent* equations. Note that this flow is different to the dissipative flows using conformal Hamiltonian mechanics studied in [12, 13], which are also Hamiltonian descent methods but employ a different dissipative force. We shall show the linear convergence of solutions of (2) to a minimum of the Hamiltonian function; first we will state a necessary assumption:

**Assumption 1.** *The Hamiltonian $\mathcal{H}$ together with a point $(x_\star, p_\star) = z_\star \in \operatorname{arg min}_z \mathcal{H}(z)$ satisfy the following:*

- $z_\star = \operatorname{arg min}_z \mathcal{H}(z)$ *is unique,*
- $\mathcal{H}(z) \geq \mathcal{H}(z_\star) = 0$ *for all $z \in \mathbb{R}^{2n}$,*

- $\mathcal{H}$ *is proper, closed, convex,*

- $\mathcal{H}$ *is continuously differentiable.*

**Theorem 1.** *If $z_t$ is following the equations of motion in (2) where $z_\star$ and the Hamiltonian function satisfy assumption 1, then the Hamiltonian converges to zero linearly (i.e., geometrically). Furthermore, $z_t$ converges to $z_\star$ and $\dot{z}_t$ converges to zero.*

*Proof.* Consider the time derivative of the Hamiltonian:

$$\dot{\mathcal{H}}(z_t) = \nabla\mathcal{H}(z_t)^T \dot{z}_t = \nabla\mathcal{H}(z_t)^T (J\nabla\mathcal{H}(z_t) + z_\star - z_t) \leq -\mathcal{H}(z_t). \tag{3}$$

since $J$ is skew-symmetric, $\mathcal{H}(z_\star) = 0$ and $\mathcal{H}$ is convex. Grönwall's inequality [18] then implies that $0 \leq \mathcal{H}(z_t) \leq \mathcal{H}(z_0)\exp(-t)$ and so $\mathcal{H}(z_t) \to 0$ linearly. Consider $M = \{z \in \mathbb{R}^{2n} : \nabla\mathcal{H}(z)^T(z_\star - z) = 0\}$. It is not too hard to see that $M = \{z_\star\}$ and that $M$ is an invariant set, since $\nabla\mathcal{H}(z'_\star)^T(z_\star - z'_\star) \geq \mathcal{H}(z)$ by convexity. Because $\mathcal{H}$ has a unique minimum, its sublevel set are bounded. Thus, we can apply Theorem 3.4 of [19] (Local Invariant Set Theorem) to argue that all solutions $z_t \to z_\star$. Further, we have $\nabla\mathcal{H}(z_t) \to 0$ by continuity and thus $\dot{z}_t \to 0$. $\qquad\square$

In contrast, consider the gradient descent flow $\dot{z}_t = -\nabla\mathcal{H}(z_t)$, which also converges since

$$\dot{\mathcal{H}}(z_t) = \nabla\mathcal{H}(z_t)^T \dot{z}_t = -\|\nabla\mathcal{H}(z_t)\|_2^2 \leq 0.$$

In this case, linear convergence is only guaranteed when $\mathcal{H}$ has some other property, such as strong convexity, which Hamiltonian descent does not require.

It may appear that these equations of motion are unrealizable without knowledge of a minimum of the Hamiltonian $z_\star$, which would defeat the goal of finding such a point. However, by a judicious choice of the Hamiltonian we can *cancel* the terms involving $z_\star$, and make the system realizable. For example, take the problem of minimizing convex $f : \mathbb{R}^n \to \mathbb{R}$, and consider the following Hamiltonian

$$\mathcal{H}(x, p) = f(x) + f^*(p) - p^T x_\star,$$

where $x_\star$ is any minimizer of $f$. Note that $(x_\star, 0) \in \text{argmin}_{(x,p)} \mathcal{H}(x, p)$. Assuming $f$ and $f^*$ are continuously differentiable and $(x_\star, 0)$ is a unique minimum of $\mathcal{H}$, then it is readily verified that this Hamiltonian satisfies assumption 1. So the solutions of the equations of motion will converge to a minimum of $\mathcal{H}$ linearly. In this case the flow is given by

$$\dot{x}_t = \nabla_p \mathcal{H}(x_t, p_t) + x_\star - x_t = \nabla f^*(p_t) - x_t$$
$$\dot{p}_t = -\nabla_x \mathcal{H}(x_t, p_t) + p_\star - p_t = -\nabla f(x_t) - p_t,$$

since $p_\star = 0$, and note that theorem 1 implies that $\dot{x}_t \to 0$, $\dot{p}_t \to 0$ and in the limit these equations reduce to the optimality condition for the problem, namely $\nabla f(x) = 0$. However, this system requires the ability to evaluate $\nabla f^*$, which is as hard as the original problem (since $x_\star = \nabla f^*(0)$). In the sequel we shall exploit the structure of composite optimization problems to avoid this requirement.

## 2.1 Affine invariance

The Hamiltonian descent equations of motion (2) are invariant to a set of affine transformations. This property is very useful since it means that the performance of an algorithm based on these equations will be much less sensitive to the conditioning of the problem than, for example, gradient descent which does not enjoy affine invariance.

To show this property, consider a non-singular matrix $M$ that satisfies $MJM^T = J$ and consider the Hamiltonian in the new coordinate system,

$$\bar{\mathcal{H}}(y) = \mathcal{H}(M^{-1}y),$$

where clearly $y_\star = Mz_\star$. At time $\tau$ we have the point $y_\tau$, and let $z_\tau = M^{-1}y_\tau$. Running Hamiltonian descent in the transformed coordinates we obtain

$$\begin{aligned}
\dot{y}_\tau &= J\nabla\bar{\mathcal{H}}(y_\tau) + y_\star - y_\tau \\
&= JM^{-T}\nabla\mathcal{H}(M^{-1}y_\tau) + Mz_\star - Mz_\tau \\
&= M(J\nabla\mathcal{H}(z_\tau) + z_\star - z_\tau) \\
&= M\dot{z}_\tau.
\end{aligned}$$

Now let $z_0 = M^{-1}y_0$, then we have $y_t = y_0 + \int_0^t \dot{y}_\tau = Mz_0 + \int_0^t M\dot{z}_\tau = Mz_t$ for all $t$, and therefore $\bar{\mathcal{H}}(y_t) = \mathcal{H}(M^{-1}Mz_t) = \mathcal{H}(z_t)$, *i.e.*, the original and transformed Hamiltonians have exactly the same value for all $t$ and thus the rate of convergence is unchanged by the transformation. The condition on $M$ is not too onerous; for example any $M$ of the form:

$$M = \begin{bmatrix} R & 0 \\ 0 & R^{-T} \end{bmatrix}$$

for nonsingular $R \in \mathbb{R}^{n \times n}$ satisfies the condition. Contrast this to vanilla gradient flow,

$$\dot{y}_\tau = -\nabla\bar{\mathcal{H}}(y_\tau) = -M^{-T}\nabla\mathcal{H}(M^{-1}y_\tau) = M^{-T}\dot{z}_\tau.$$

Again setting $z_0 = M^{-1}y_0$ we obtain $y_t = y_0 + \int_0^t \dot{y}_\tau = Mz_0 + \int_0^t M^{-T}\dot{z}_\tau \neq Mz_t$ except in the case that $M^T M = I$, *i.e.*, $M$ is orthogonal.

## 2.2 Discretizations

There are many possible ways to discretize the Hamiltonian descent equations, see, *e.g.*, [20]. Here we present two simple approaches and prove their convergence under certain conditions. Later we shall show that other discretizations correspond to already known algorithms.

### 2.2.1 Implicit

Consider the following implicit discretization of (2), for some $\epsilon > 0$ we take

$$z^{k+1} = z^k + \epsilon(J\nabla\mathcal{H}(z^{k+1}) + z_\star - z^{k+1}). \tag{4}$$

Consider the change in Hamiltonian value at iteration $k$, $\Delta_k = \mathcal{H}(z^{k+1}) - \mathcal{H}(z^k)$:

$$\Delta_k \leq \nabla\mathcal{H}(z^{k+1})^T(z^{k+1} - z^k) = \epsilon\nabla\mathcal{H}(z^{k+1})^T(J\nabla\mathcal{H}(z^{k+1}) + z_\star - z^{k+1}) \leq -\epsilon\mathcal{H}(z^{k+1})$$

since $J$ is skew-symmetric, $\mathcal{H}(z_\star) = 0$ and $\mathcal{H}$ is convex. From this we have $\mathcal{H}(z^k) \leq (1+\epsilon)^{-k}\mathcal{H}(z_0)$. Thus the implicit discretization exhibits linear convergence in discrete-time, without restriction on the step-size $\epsilon$. However, this scheme is very difficult to implement in practice, since it requires solving a non-linear equation for $z^{k+1}$ at every step.

### 2.2.2 Explicit

Now consider the explicit discretization

$$z^{k+1} = z^k + \epsilon(J\nabla\mathcal{H}(z^k) + z_\star - z^k), \tag{5}$$

this differs from the implicit discretization in that the right hand side depends solely on $z^k$ rather than $z^{k+1}$, and therefore is much more practical to implement. If we assume that the gradient of $\mathcal{H}$ is $L$-Lipschitz, then we can show that this sequence converges and that the Hamiltonian converges to zero like $O(1/k)$. The proof of this result is included in the appendix. If, in addition, $\mathcal{H}$ is $\mu > 0$ strongly convex, then we can show that the Hamiltonian converges to zero like $O(\lambda^k)$ for some $\lambda < 1$. The proof of this result, along the explicit dependence of $\lambda$ on $L$ and $\mu$ is given in the appendix.

We must mention here that both proofs are somewhat lacking. For example, under the assumptions of $L$-Lipschitzness of $\nabla\mathcal{H}$ and $\mu$ strong convexity of $\mathcal{H}$, our analysis requires that the step-size $\epsilon$ depend on both $L$ and $\mu$. This is a stronger requirement than the classical gradient descent analysis. Moreover, the rate $\lambda$ scales poorly with the condition number $L/\mu$ as compared to gradient descent. This may be due to the fact that both analyses depend strongly on the values of $L$ or $\mu$, which are not invariant to affine transformation even though the equations of motion are. We suspect that a tighter analysis is possible under assumptions whose structure mirror the affine invariance structure of the dynamics.

## 3 Composite optimization

Now we come to the main problem we investigate in this paper. Consider a convex optimization problem consisting of the sum of two convex, closed, proper functions $h : \mathbb{R}^n \to \mathbb{R}$ and $g : \mathbb{R}^m \to \mathbb{R}$:

$$\text{minimize} \quad f(y) := h(Ay) + g(y) \tag{6}$$

over variable $y \in \mathbb{R}^m$, with data matrix $A \in \mathbb{R}^{n \times m}$. This problem is sometimes referred to as a composite optimization problem, see, *e.g.*, [21]. The dual problem is given by

$$\text{maximize} \quad d(p) := -h^*(-p) - g^*(A^T p), \tag{7}$$

over $p \in \mathbb{R}^n$. We assume that $h$ and $g^*$ are both differentiable, which will help ensure that the Hamiltonian we derive satisfies assumption 1. Weak duality tells us that for any $y, p$ we have $f(y) \geq d(p)$, with equality if and only if $y$ and $p$ are primal-dual optimal, since strong duality always holds for this problem (under mild technical conditions [22, §5.2.3]). We can rewrite the primal and dual problems in equality constrained form:

$$
\begin{array}{llll}
\text{minimize} & h(x) + g(y) & \text{maximize} & -h^*(-p) - g^*(q) \\
\text{subject to} & x = Ay, & \text{subject to} & q = A^T p,
\end{array} \tag{8}
$$

and obtain necessary and sufficient optimality conditions in terms of all four variables:

$$
\begin{aligned}
\nabla g^*(q_\star) - y_\star &= 0 \\
Ay_\star - x_\star &= 0 \\
-\nabla h(x_\star) - p_\star &= 0 \\
A^T p_\star - q_\star &= 0,
\end{aligned} \tag{9}
$$

the proof of which is included in the appendix.

## 3.1 Duality gap as Hamiltonian

In this section we derive a partial duality gap for problem (8) and use it as our Hamiltonian function to derive equations of motion. Then we shall show that in the limit the equations we derive satisfy the conditions necessary and sufficient for optimality (9). We start by introducing dual variable $p$ for the equality constraint in the primal problem (8) to obtain $h(x) + g(y) + p^T(x - Ay)$, and taking the Legendre transform of $g$ we get the 'full' Lagrangian in terms of all four primal and dual variables:

$$\mathcal{L}(x, y, p, q) = h(x) - g^*(q) + y^T q + p^T(x - Ay),$$

which is convex-concave in $(x, y)$ and $(p, q)$. We refer to this as the full Lagrangian, because if we maximize over $(p, q)$ we recover the primal problem in (8) and if we minimize over $(x, y)$ we recover the dual problem in (8). Denote by $(y_\star, p_\star)$ any primal-dual optimal point and let $x_\star = Ay_\star$, $q_\star = A^T p_\star$, and $f_\star = f(y_\star) = d(p_\star)$, then a simple calculation yields

$$\mathcal{L}(x_\star, y_\star, p, q) \leq \max_{p,q} \mathcal{L}(x_\star, y_\star, p, q) = f_\star = \min_{x,y} \mathcal{L}(x, y, p_\star, q_\star) \leq \mathcal{L}(x, y, p_\star, q_\star).$$

This is due to strong duality holding for this problem. In other words, if we substitute in the optimal primal or dual variables into the Lagrangian, then we obtain valid lower and upper bounds respectively. Then maximizing and minimizing these bounds over the remaining variables yields the optimal objective value, $f_\star$. Thus, the difference between these two functions is a partial *duality gap* (though uncomputable without knowledge of a primal-dual optimal point),

$$
\begin{aligned}
\text{gap}(x, q) &= \mathcal{L}(x, y, p_\star, q_\star) - \mathcal{L}(x_\star, y_\star, p, q) \\
&= h(x) - h(x_\star) + g^*(q) - g^*(q_\star) + x^T p_\star - q^T y_\star \\
&\geq 0,
\end{aligned} \tag{10}
$$

with equality only when the Lagrangians are equal, *i.e.*, are optimal. Note that the gap only depends on $x, q$, because the effect of $y$ and $p$ is cancelled out. This gap can also be written in terms of Bregman divergences, where the Bregman divergence between points $u$ and $v$ induced by a differentiable convex function $h$ is defined as $D_h(u, v) = h(u) - h(v) - \nabla h(v)^T(u - v)$, which is always nonnegative due the convexity of $h$. Though not a true distance metric, it does have some useful 'distance-like' properties [23, 24]. We show in the appendix that our partial duality gap can be rewritten as

$$\text{gap}(x, q) = D_h(x, x_\star) + D_{g^*}(q, q_\star).$$

In other words, the gap also corresponds to a 'distance' between the current iterates and their optimal values, as induced by the functions $h$ and $g^*$. Furthermore, we show in the appendix that this partial duality gap is a lower bound on the full duality gap, *i.e.*,

$$f(y) - d(p) \geq \text{gap}(Ay, A^T p).$$

The gap is not in the form of a Hamiltonian, since the variable $x$ and $q$ are of different dimension. We can reparameterize $q = A^T p$ or $x = Ay$, which yields two possible Hamiltonians, one in dimension $n$ and one in dimension $m$. The first of which is

$$\mathcal{H}(x, p) = \text{gap}(x, A^T p) = h(x) - h(x_\star) + g^*(A^T p) - g^*(A^T p_\star) + x^T p_\star - p^T x_\star. \tag{11}$$

Due to the assumptions on $h$ and $g^*$ we know that $\mathcal{H}$ is convex and differentiable, and evidently $\mathcal{H}(x, p) \geq \mathcal{H}(x_\star, p_\star) = 0$. This Hamiltonian function combined with the equations of motion in equation (2) yields dynamics

$$
\begin{aligned}
\dot{x}_t &= \nabla_p \mathcal{H}(x_t, p_t) + x_\star - x_t = A\nabla g^*(A^T p_t) - x_t \\
\dot{p}_t &= -\nabla_x \mathcal{H}(x_t, p_t) + p_\star - p_t = -\nabla h(x_t) - p_t.
\end{aligned}
\tag{12}
$$

We could rewrite these equations as

$$
\begin{aligned}
\nabla g^*(q_t) - y_t &= 0 \\
Ay_t - x_t &= \dot{x}_t \\
-\nabla h(x_t) - p_t &= \dot{p}_t \\
A^T p_t - q_t &= 0,
\end{aligned}
$$

If $\dot{x}_t \to 0$ and $\dot{p}_t \to 0$, then the above equations converge to the conditions necessary and sufficient for optimality, as given in equation (9). This convergence could be guaranteed by theorem 1, when $\mathcal{H}$ has a unique minimum (and thus satisfies all of assumption 1). Still, we suspect it is possible to prove the convergence of the system without this requirement on $\mathcal{H}$'s minima.

The second Hamiltonian is given by

$$\mathcal{H}(y, q) = \text{gap}(Ay, q) = h(Ay) - h(Ay_\star) + g^*(q) - g^*(q_\star) + y^T q_\star - q^T y_\star \tag{13}$$

which yields equations of motion

$$
\begin{aligned}
\dot{y}_t &= \nabla_q \mathcal{H}(y_t, q_t) + y_\star - y_t = \nabla g^*(q_t) - y_t \\
\dot{q}_t &= -\nabla_y \mathcal{H}(y_t, q_t) + q_\star - q_t = -A^T \nabla h(Ay_t) - q_t,
\end{aligned}
\tag{14}
$$

or equivalently

$$
\begin{aligned}
\nabla g^*(q_t) - y_t &= \dot{y}_t \\
Ay_t - x_t &= 0 \\
-\nabla h(x_t) - p_t &= 0 \\
A^T p_t - q_t &= \dot{q}_t.
\end{aligned}
$$

Again, if $\dot{y}_t \to 0$ and $\dot{q}_t \to 0$, this system will also satisfy the optimality conditions of (9). Finally, theorem 1 implies that both of these ODEs exhibit linear convergence of the Hamiltonian, *i.e.*, linear convergence of the partial duality gap (10), to zero.

## 4   Connection to other methods

### 4.1   ADMM

In this section we show how a particular discretization of our ODE yields the well-known Alternating direction method of multipliers algorithm (ADMM) [25, 26] when applied to problem (6). We should note that in related work the authors of [27] derive a different ODE that when discretized also yields ADMM, as well as a related ODE that corresponds to accelerated ADMM [28]. There is no contradiction here since many ODEs can correspond to the same procedure when discretized.

In order to prove that ADMM is equivalent to a discretization of Hamiltonian descent we will require the generalized Moreau decomposition, which we present next. In the statement of the lemma we use the notation $(A\partial f A^T)$ to represent the multi-valued operator defined as $(A\partial f A^T)(x) = A(\partial f(A^T x)) = \{Az \mid z \in \partial f(A^T x)\}$.

**Lemma 1.** *For convex, closed, proper function $f : \mathbb{R}^m \to \mathbb{R}$ and matrix $A \in \mathbb{R}^{n \times m}$, any point $x \in \mathbb{R}^n$ satisfies*

$$x = (I + \rho A\partial f A^T)^{-1} x + \rho A(\partial f^* + \rho A^T A)^{-1} A^T x.$$

We defer the proof to the appendix. To derive ADMM we employ a standard trick in discretizing differential equations: We add and subtract a term to the dynamics which we shall discretize at different points, which in the limit of infinitesimal step size will vanish, recovering the original ODE. Starting from equation (12) and for any $\rho > 0$ the modified ODE is

$$\dot{p}_t = -\nabla h(x_t) - p_t - \rho(x_t - x_t)$$
$$\dot{x}_t = A\nabla g^*(A^T p_t) - x_t + (1/\rho)(p_t - p_t).$$

Now we discretize as follows:

$$(p^k - p^{k-1})/\epsilon = -\nabla h(x^{k+1}) - p^k - \rho(x^{k+1} - x^k)$$
$$(x^{k+1} - x^k)/\epsilon = A\nabla g^*(A^T p^{k+1}) - x^k + (1/\rho)(p^{k+1} - p^k).$$

Setting $\epsilon = 1$ yields

$$x^{k+1} = (\rho I + \nabla h)^{-1}(\rho x^k - 2p^k + p^{k-1})$$
$$p^{k+1} = (I + \rho A\nabla g^* A^T)^{-1}(p^k + \rho x^{k+1})$$
$$= p^k + \rho x^{k+1} - \rho A(\partial g + \rho A^T A)^{-1}A^T(p^k + \rho x^{k+1})$$
$$= p^k + \rho x^{k+1} - \rho A y^{k+1}$$

where we used the generalized Moreau decomposition and introduced variable sequence $y^k \in \mathbb{R}^m$, and note that from the last equation we have that $\rho x^k - 2p^k + p^{k-1} = \rho A y^k - p^k$. Finally this brings us to ADMM; from any initial $y^0, p^0$ iterate

$$x^{k+1} = (\rho I + \nabla h)^{-1}(\rho A y^k - p^k)$$
$$y^{k+1} \in (\rho A^T A + \partial g)^{-1}A^T(p^k + \rho x^{k+1})$$
$$p^{k+1} = p^k + \rho(x^{k+1} - A y^{k+1}).$$

Evidently we have lost the affine invariance property of our ODE. However we might expect ADMM to be somewhat more robust to conditioning than gradient descent, which appears to be true empirically [25].

## 4.2  PDHG

The primal-dual hybrid gradient technique (PDHG), also called Chambolle-Pock, is another operator splitting technique with a slightly different form to ADMM. In particular, PDHG only requires multiplies with $A$ and $A^T$ rather than requiring $A$ in the proximal step [29, 30, 31]. When applied to problem (6) PDHG yields the following iterates

$$p^{k+1} = -(I + \rho\partial h^*)^{-1}(\rho A y^k - p^k)$$
$$y^{k+1} = (I + \sigma\partial g)^{-1}(\sigma A^T p^{k+1} + y^k).$$

In the appendix we show that this corresponds to a particular discretization of Hamiltonian descent, with step size $\epsilon = 1$. Note that the sign of the dual variable $p^k$ is different when compared to [31], this is due to the fact that the dual problem they consider negates the dual variable when compared to ours, so this is fixed by rewriting the iterations in terms of $-p^k$.

## 5  Numerical experiments

In this section we present two numerical examples where we compare the explicit discretization of Hamiltonian descent flow to gradient descent. Due to the affine invariance property of Hamiltonian descent we expect our technique to outperform when the conditioning of the problem is poor, so we generate examples with bad conditioning to test that.

### 5.1  Regularized least-squares

Consider the following $\ell_2$-regularized least-squares problem

$$\text{minimize} \quad (1/2)\|Ay - b\|_2^2 + (\lambda/2)\|By\|_2^2, \tag{15}$$

over variable $y \in \mathbb{R}^m$, where $A \in \mathbb{R}^{n \times m}$, $B \in \mathbb{R}^{m \times m}$, and $\lambda \geq 0$ are data. In the notation of problem (6) we let $h(x) = (1/2)\|x - b\|_2^2$ and $g(y) = \lambda\|By\|_2^2$, and so $\nabla g^*(q) = \operatorname{argmax}_y(y^T q - \lambda\|By\|_2^2)$ which we assume is always well-defined (*i.e.*, $B^T B$ is invertible). We apply the explicit discretization (5) of the dynamics given in equation (14) to this problem. To demonstrate the practical effect of affine invariance, we randomly generate a nonsingular matrix $M$ and solve a sequence of optimization problems where $A$ is replaced with $\hat{A}_j = AM^j$ and $B$ is replaced with $\hat{B}_j = BM^j$ for $j = 0, 1, \ldots, j^{\max}$. Note that the optimal objective value of this perturbed problem is unchanged from the original, and the solution for each perturbed problem can be obtained by $(\hat{y}_\star)_j = M^{-j}y_\star$, where $y_\star$ solves the original problem (*i.e.*, with $j = 0$). However, the conditioning of the problem is changed - $M$ is selected so that the conditioning of the data is worsening for increasing $j$. We compare our algorithm to vanilla gradient descent, to proximal gradient descent [32] (where the prox-step is on the $g$ term so it is of a similar cost to our method), and to restarted accelerated gradient descent [6, 33], and observe the effect of the worsening conditioning.

We chose $m = n = 1000$ and for simplicity we chose $B = I$, $\lambda = 1$, and randomly generated each entry in $A$ to be IID $\mathcal{N}(0, 1)$. The best step size was chosen via exhaustive search for all three algorithms. The matrix $M$ was randomly generated but chosen in such a way so as to be close to the identity. For $j = 0$ the condition number of the matrix $\hat{A}_j^T \hat{A}_j + \lambda \hat{B}_j^T \hat{B}_j$ was $4.0 \times 10^3$, and for $j = j^{\max} = 20$ the condition number had grown to $2.2 \times 10^{14}$, a dramatic increase. Figure (1a) shows the performance of both our technique and gradient descent on this sequence of problems. The gradient descent traces are in orange, with a different trace for each $j$. The fastest converging trace corresponds to $j = 0$, the best conditioned problem. As the conditioning deteriorates the convergence is impacted, getting slower with each increase in $j$. In the appendix we additionally include Figure 3 which compares our technique to proximal gradient, restarted accelerated gradient, and conjugate gradient. All three additional techniques display the same deterioration as the conditioning worsens. By problem $j = 20$ no variant of gradient descent or conjugate gradient has reduced the primal objective error, defined as $\min_k(f(y^k) - f_\star)$, to under $O(100)$. By contrast, our technique is completely *unaffected* by the changing data, with every trace essentially identical (up to some numerical tolerances). Furthermore, we used the exact same step size for every run of our method. This is because the discretization procedure preserved the affine invariance of the continuous ODE it is approximating, so the changing conditioning of the data has no effect. In Figure (1b) we plot the Hamiltonian (13) (*i.e.*, the partial duality gap) and the full duality gap: $f(y^k) - d(p^k)$, for Hamiltonian descent for each value of $j$. Once again the traces lie directly on top of each other, until numerical errors start to have an impact. We note that the Hamiltonian decreases at each iteration, and converges linearly. The duality gap and the objective values do not necessarily decrease at each iteration, but do appear to enjoy linear convergence for each $j$.

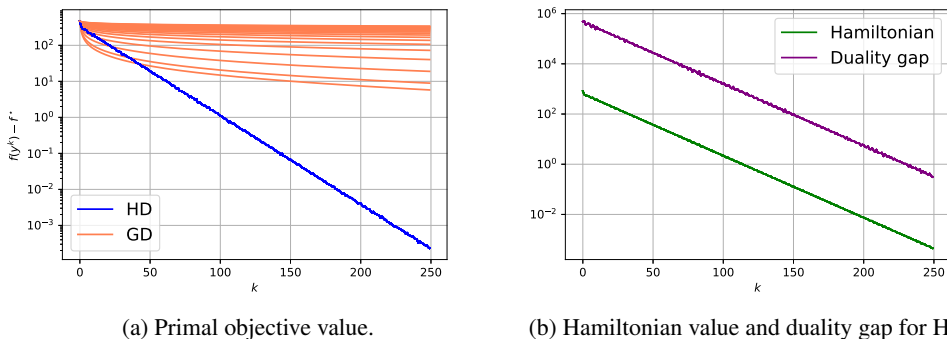

(a) Primal objective value.

(b) Hamiltonian value and duality gap for HD.

Figure 1: Comparison of Hamiltonian descent (HD) and Gradient descent (GD) for problem (15).

## 5.2 Elastic net regularized logistic regression

In logistic regression the goal is to learn a classifier to separate a set of data points based on their labels, which we take to be either $1$ or $-1$. The elastic net is a type of regularization that promotes sparsity and small weights in the solution [34]. Given data points $a_i \in \mathbb{R}^m$ with corresponding label

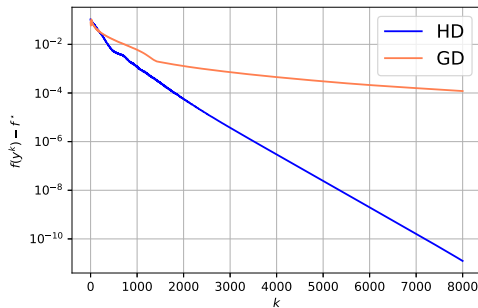

Figure 2: Comparison of Hamiltonian descent (HD) and Gradient descent (GD) for problem (16).

$l_i \in \{-1, 1\}$ for $i = 1, \ldots, n$, the elastic net regularized logistic regression problem is given by

$$\text{minimize} \quad (1/n) \sum_{i=1}^{n} \log(1 + \exp(l_i a_i^T y)) + \lambda_1 \|y\|_1 + (\lambda_2/2)\|y\|_2^2 \tag{16}$$

over the variable $y \in \mathbb{R}^m$, where $\lambda_1 \geq 0$, and $\lambda_2 \geq 0$ control the strength of the regularization. In the notation of problem (6) we take $h(x) = (1/n) \sum_{i=1}^{n} \log(1 + \exp(l_i x_i))$ and $g(y) = \lambda_1 \|y\|_1 + (\lambda_2/2)\|y\|_2^2$. We have a closed form expression for the gradient of $g^*$ given by the soft-thresholding operator:

$$(\nabla g^*(q))_i = (1/\lambda_2) \begin{cases} q_i - \lambda_1 & q_i \geq \lambda_1 \\ 0 & |q_i| \leq \lambda_1 \\ q_i + \lambda_1 & q_i \leq -\lambda_1. \end{cases}$$

We compare the explicit discretization (5) of Hamiltonian descent in equation (14) to proximal gradient descent [32], which in this case has the exact same per-iteration cost since it also relies on taking the gradient of $h$ and applying the soft-thresholding operator. We chose dimension $m = 500$ and $n = 1000$ data points and we set $\lambda_1 = \lambda_2 = 0.01$. The data were generated randomly, and then perturbed so as to give a high condition number, which was $1.0 \times 10^8$. The best step size for both algorithms was found using exhaustive search. In Figure 2 we show the primal objective value error for both algorithms, where the true solution was found using convex cone solver SCS [35, 36]. Hamiltonian descent dramatically outperforms gradient descent on this problem, despite having the same per-iteration cost. This is unsurprising because we would expect Hamiltonian descent to be less sensitive to the poor conditioning of the data, due to the affine invariance property.

## 6 Conclusion

Starting from Hamiltonian mechanics in classical physics, we derived a Hamiltonian descent continuous ODE that converges linearly to a minimum of the Hamiltonian function. We applied Hamiltonian descent to a convex composite optimization problem, and proved linear convergence of the duality gap, a measure of how far from optimal a primal-dual point is. In some sense applying Hamiltonian descent to this problem is natural, since we can identify one of the terms in the objective as being the 'potential' energy and the other as the 'kinetic' energy. We provided two discretizations that are guaranteed to converge to the optimum under certain assumptions, and also demonstrated that some well-known algorithms correspond to other discretizations of our ODE. In particular we show that a particular discretization yields ADMM. We conclude with two numerical examples that show our method is much more robust to numerical conditioning than standard gradient methods.

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
