[Supplementary Material]

## Appendix

**Necessary and sufficient conditions for optimality of (8)**

Recall that the primal-dual problems we are considering are:

$$\begin{array}{llll}
\text{minimize} & h(x) + g(y) & \text{maximize} & -h^*(-p) - g^*(q) \\
\text{subject to} & x = Ay, & \text{subject to} & q = A^T p,
\end{array}$$

over primal variables $y \in \mathbb{R}^m$, $x \in \mathbb{R}^n$, dual variables $p \in \mathbb{R}^n$, $q \in \mathbb{R}^m$, where matrix $A \in \mathbb{R}^{n \times m}$ is data and the functions $h : \mathbb{R}^n \to \mathbb{R} \cup \{\infty\}$ and $g : \mathbb{R}^m \to \mathbb{R} \cup \{\infty\}$ are convex, with convex conjugates $h^*$ and $g^*$ respectively. The necessary and sufficient condition for optimality of $y_\star$ for the primal problem is

$$0 \in A^T \partial h(Ay_\star) + \partial g(y_\star).$$

We can rewrite the optimality condition as, $\exists p_\star \in -\partial h(Ay_\star)$,

$$0 \in -A^T p_\star + \partial g(y_\star). \tag{17}$$

Note we have the following property for any proper convex $f$ (see Theorem 23.5 of [17]). For any $x^*, x \in \mathbb{R}^m$,

$$x \in \partial f^*(x^*) \qquad \text{iff} \qquad x^* \in \partial f(x) \tag{18}$$

Now using this fact, for any $p_\star$ satisfying (17) we get the following two inclusions

$$\begin{aligned}
Ay_\star &\in A\partial g^*(A^T p_\star) \\
Ay_\star &\in \partial h^*(-p_\star)
\end{aligned} \tag{19}$$

Together, this implies

$$0 \in -\partial h^*(-p_\star) + A\partial g^*(A^T p_\star),$$

which is the necessary and sufficient condition for $p_\star$ to be optimal for the dual problem. Then taking primal-dual optimal $(y_\star, p_\star)$ and introducing $x_\star = Ay_\star$ and $q_\star = A^T p_\star$, we get

$$\begin{aligned}
y_\star &\in \partial g^*(q_\star) \\
x_\star &= Ay_\star \\
-p_\star &\in \partial h(x_\star) \\
q_\star &= A^T p_\star,
\end{aligned}$$

which are the necessary and sufficient conditions for $(x_\star, y_\star, p_\star, q_\star)$ to be primal-dual optimal for (8). In the main text we assumed that $h$ and $g^*$ were differentiable, in which case we can replace subdifferentials with gradients, and inclusion with equality.

**Relationship between the duality gap and Bregman divergences**

Starting with the definition of Bregman divergences and noting that $\nabla h(x_\star) = -p_\star$,

$$D_h(x, x_\star) = h(x) - h(x_\star) + p_\star^T(x - x_\star),$$

and similarly using $\nabla g^*(q_\star) = y_\star$,

$$D_{g^*}(q, q_\star) = g^*(q) - g^*(q_\star) - y_\star^T(q - q_\star).$$

Using $p_\star^T x_\star = p_\star^T(Ay_\star) = q_\star^T y_\star$ and summing the two Bregman divergences yields the gap (10).

Now let us define

$$\hat{D}_{h^*}(-p, -p_\star) = h^*(-p) - h^*(-p_\star) - x_\star^T(-p + p_\star),$$

and

$$\hat{D}_g(y, y_\star) = g(y) - g(y_\star) - q_\star^T(y - y_\star),$$

which are both nonnegative due to the convexity of $h^*$ and $g$, and note that if $h^*$ and $g$ are differentiable then these are just Bregman divergences, in which case we could drop the 'hat' notation.

Now we shall show that the usual duality gap can be decomposed into the sum of four (pseudo)-Bregman divergences. Let $\hat{D}_f(y, y_\star) = f(y) - f(y_\star)$, which by the linearity of (pseudo)-Bregman divergences satisifes

$$\hat{D}_f(y, y_\star) = \hat{D}_{h \circ A + g}(y, y_\star) = D_h(Ay, x_\star) + \hat{D}_g(y, y_\star)$$

and similarly denoting $\hat{D}_d(p, p_\star) = -d(p) + d(p_\star)$ we have

$$\hat{D}_d(p, p_\star) = \hat{D}_{h^*}(-p, -p_\star) + D_{g^*}(A^T p, q_\star).$$

Summing these and using the fact that strong duality implies that $f(y_\star) = d(p_\star)$ we obtain

$$f(y) - d(p) = D_h(Ay, x_\star) + \hat{D}_g(y, y_\star) + \hat{D}_{h^*}(-p, -p_\star) + D_{g^*}(A^T p, q_\star),$$

which, due to the nonnegativity of $\hat{D}_{h^*}$ and $\hat{D}_g$, implies that

$$f(y) - d(p) \geq D_h(Ay, x_\star) + D_{g^*}(A^T p, q_\star) = \mathrm{gap}(Ay, A^T p).$$

**Proof of generalized Moreau decomposition**

**Lemma 1.** *Given a convex, closed, proper function $f : \mathbb{R}^m \to \mathbb{R}$, matrix $A \in \mathbb{R}^{n \times m}$, and $\rho > 0$. Any point $x \in \mathbb{R}^n$ satisfies*

$$x = (I + \rho A \partial f A^T)^{-1} x + \rho A (\partial f^* + \rho A^T A)^{-1} A^T x.$$

*Proof.* Recall that $(I + \rho A \partial f A^T)^{-1}$ is always single-valued, because it is the proximal operator of the convex function $f \circ A^T$ [32]. So to start we shall show that $A(\partial f^* + \rho A^T A)^{-1} q$ is also single-valued for any $q$, $A$, and convex $f^*$. Choose $y$ and $z$ to be any two elements of $(\partial f^* + \rho A^T A)^{-1} q$. We shall show that it must be the case that $Ay = Az$, even if $z \neq y$. Membership of the set implies that

$$q - \rho A^T A y \in \partial f^*(y)$$
$$q - \rho A^T A z \in \partial f^*(z),$$

therefore by convexity and the definition of subdifferentials we have

$$f^*(z) \geq f^*(y) + (q - \rho A^T A y)^T (z - y)$$
$$f^*(y) \geq f^*(z) + (q - \rho A^T A z)^T (y - z),$$

and adding these we get

$$\begin{aligned} 0 &\geq (q - \rho A^T A y)^T (z - y) + (q - \rho A^T A z)^T (y - z) \\ &= \rho (A^T A y)^T (y - z) - \rho (A^T A z)^T (y - z) \\ &= \rho (y - z)^T A^T A (y - z) \\ &= \rho \|A(y - z)\|_2^2, \end{aligned}$$

which implies that $Az = Ay$, so $A(\partial f^* + \rho A^T A)^{-1} q$ must be single-valued.

Now let $y = (\partial f^* + \rho A^T A)^{-1} A^T x$ (which is valid, because it is single-valued). We will make use of the following fact for any proper convex $f$ (see Theorem 23.5 of [17]). For any $x^*, x \in \mathbb{R}^m$,

$$x \in \partial f^*(x^*) \qquad \text{iff} \qquad x^* \in \partial f(x) \qquad (20)$$

Now using (20),

$$\begin{aligned} A^T(x - \rho A y) \in \partial f^*(y) &\implies y \in (\partial f A^T)(x - \rho A y) \\ &\implies \rho A y \in \rho (A \partial f A^T)(x - \rho A y) \\ &\implies x \in (I + \rho A \partial f A^T)(x - \rho A y) \end{aligned}$$

Since $(I + \rho A \partial f A^T)^{-1} x$ is single valued, we can use (20) along with the definition of $y$ to finish the proof:

$$\begin{aligned} x &= (I + \rho A \partial f A^T)^{-1} x + \rho A y \\ &= (I + \rho A \partial f A^T)^{-1} x + \rho A (\partial f^* + \rho A^T A)^{-1} A^T x. \end{aligned}$$

$\square$

The Moreau decomposition can be seen as a generalization of an orthogonal decomposition induced by a subspace, and the standard statement of the theorem assumes that $A = I$, see, *e.g.*, [32]. This extension can be interpreted as a decomposition when the projection is weighted by the matrix $A$, since

$$\operatorname*{argmin}_{v} \left( f(A^T v) + (1/2)\|v - y\|_2^2 \right) = (I + A\partial f A^T)^{-1} y$$

$$\operatorname*{argmin}_{u} \left( f^*(u) + (1/2)\|Au - x\|_2^2 \right) = (\partial f^* + A^T A)^{-1} A^T x.$$

**Convergence of Explicit discretization scheme when $\nabla\mathcal{H}$ is $L$-Lipschitz**

To show convergence of the scheme presented in equation (5) we shall use the additional assumption that $\mathcal{H}$ has an $L$-Lipschitz gradient, which implies that

$$\mathcal{H}(v) \geq \mathcal{H}(u) + \nabla\mathcal{H}(u)^T (v - u) + (1/2L)\|\nabla\mathcal{H}(v) - \nabla\mathcal{H}(u)\|_2^2$$

$$\mathcal{H}(v) \leq \mathcal{H}(u) + \nabla\mathcal{H}(u)^T (v - u) + (L/2)\|v - u\|_2^2,$$

for any $u, v$. Using this we can write:

$$\begin{aligned}
\mathcal{H}(z^{k+1}) - \mathcal{H}(z^k) &\leq \nabla\mathcal{H}(z^k)^T (z^{k+1} - z^k) + (L/2)\|z^{k+1} - z^k\|_2^2 \\
&= \epsilon\nabla\mathcal{H}(z^k)^T (J\nabla\mathcal{H}(z^k) + z_\star - z^k) + (\epsilon^2 L/2)\|J\nabla\mathcal{H}(z^k) + z_\star - z^k\|_2^2 \quad (21) \\
&\leq -\epsilon\mathcal{H}(z^k) - (\epsilon/2L)\|\nabla\mathcal{H}(z^k)\|_2^2 + (\epsilon^2 L/2)\|J\nabla\mathcal{H}(z^k) + z_\star - z^k\|_2^2,
\end{aligned}$$

where the first inequality is a consequence of the Lipschitz assumption, and the last is a combination of the Lipschitz assumption and the fact that $J$ is skew symmetric. Now we will use the following identity:

$$\|(1 - \epsilon)u + \epsilon v\|_2^2 = (1 - \epsilon)\|u\|_2^2 + \epsilon\|v\|_2^2 - \epsilon(1 - \epsilon)\|u - v\|_2^2$$

for any $u, v$ and $\epsilon \in \mathbb{R}$. We apply this to the following

$$\|z^{k+1} - z_\star\|_2^2 = (1 - \epsilon)\|z^k - z_\star\|_2^2 + \epsilon\|\nabla\mathcal{H}(z^k)\|_2^2 - \epsilon(1 - \epsilon)\|J\nabla\mathcal{H}(z^k) + z_\star - z^k\|_2^2$$

where we used the fact that $\|J\nabla\mathcal{H}(z)\|_2^2 = \|\nabla\mathcal{H}(z)\|_2^2$ since $J^T J = I$. This allows us to replace the last term in (21)

$$\begin{aligned}
\mathcal{H}(z^{k+1}) - \mathcal{H}(z^k) \leq {}&-\epsilon\mathcal{H}(z^k) - (\epsilon/2L)\|\nabla\mathcal{H}(z^k)\|_2^2 + \\
&\frac{\epsilon L}{2(1 - \epsilon)} \left( (1 - \epsilon)\|z^k - z_\star\|_2^2 + \epsilon\|\nabla\mathcal{H}(z^k)\|_2^2 - \|z^{k+1} - z_\star\|_2^2 \right).
\end{aligned} \quad (22)$$

Now select $\epsilon$ to satisfy

$$\frac{\epsilon}{2L} \geq \frac{\epsilon^2 L}{2(1 - \epsilon)},$$

which removes the terms involving $\|\nabla\mathcal{H}(z^k)\|_2^2$. For simplicity we shall take $\epsilon = 1/(L^2 + 1)$, which satisfies the condition. Now we take the sum of (22), which telescopes to yield

$$\mathcal{H}(z^T) - \mathcal{H}(z^0) \leq -\epsilon\sum_{k=0}^{T-1} \mathcal{H}(z^k) + (2L)^{-1}(\|z^0 - z_\star\|_2^2 - \|z^T - z_\star\|_2^2). \quad (23)$$

Now consider the averaged iterate $\bar{z}^T = (1/T)\sum_{k=0}^{T-1} z^k$

$$\mathcal{H}(\bar{z}^T) \leq \frac{1}{T}\sum_{k=0}^{T-1} \mathcal{H}(z^k) \leq \frac{1}{\epsilon T} \left( \mathcal{H}(z^0) + (2L)^{-1}\|z^0 - z_\star\|_2^2 \right),$$

where the first inequality is Jensen's, and the second follows from (23) and the nonnegativity of $\mathcal{H}$. In other words $\mathcal{H}(\bar{z}^k) \to 0$, and the rate of convergence is $O(1/k)$.

**Convergence of Explicit discretization scheme when $\nabla\mathcal{H}$ is $L$-Lipschitz and $\mathcal{H}$ is $\mu$ strongly convex**

Here we show the convergence of the scheme presented in equation (5) under the assumption that $\mathcal{H}$ has an $L$-Lipschitz gradient and is $\mu$ strongly convex for $L \geq \mu > 0$. We show that the $\mathcal{H}(z^k)$ converges linearly.

The assumption of $L$-Lipschitz gradients implies,

$$\mathcal{H}(v) \geq \mathcal{H}(u) + \nabla\mathcal{H}(u)^T(v - u) + (1/L)\|\nabla\mathcal{H}(v) - \nabla\mathcal{H}(u)\|_2^2/2$$
$$\mathcal{H}(v) \leq \mathcal{H}(u) + \nabla\mathcal{H}(u)^T(v - u) + L\|v - u\|_2^2/2,$$

for any $u, v$. The $\mu$ strong convexity assumption implies

$$\mathcal{H}(v) \leq \mathcal{H}(u) + \nabla\mathcal{H}(u)^T(v - u) + (1/\mu)\|\nabla\mathcal{H}(v) - \nabla\mathcal{H}(u)\|_2^2/2$$
$$\mathcal{H}(v) \geq \mathcal{H}(u) + \nabla\mathcal{H}(u)^T(v - u) + (\mu)\|v - u\|_2^2/2,$$

for any $u, v$. In particular, we use

$$\mu\mathcal{H}(z) \leq \|\nabla\mathcal{H}(z)\|_2^2/2 \leq L\mathcal{H}(z)$$
$$\mu\|z - z_\star\|_2^2/2 \leq \mathcal{H}(z) \leq L\|z - z_\star\|_2^2/2$$

Using this we can write:

$$\begin{aligned}
\mathcal{H}(z^{k+1}) - \mathcal{H}(z^k) &\leq \nabla\mathcal{H}(z^k)^T(z^{k+1} - z^k) + (L/2)\|z^{k+1} - z^k\|_2^2 \\
&= \epsilon\nabla\mathcal{H}(z^k)^T(J\nabla\mathcal{H}(z^k) + z_\star - z^k) + (\epsilon^2 L/2)\|J\nabla\mathcal{H}(z^k) + z_\star - z^k\|_2^2 \\
&\leq -\epsilon\mathcal{H}(z^k) - (\epsilon/2L)\|\nabla\mathcal{H}(z^k)\|_2^2 + (\epsilon^2 L/2)\|J\nabla\mathcal{H}(z^k) + z_\star - z^k\|_2^2, \\
&\leq -\epsilon\left(1 + \frac{\mu}{L}\right)\mathcal{H}(z^k) + (\epsilon^2 L/2)\|J\nabla\mathcal{H}(z^k) + z_\star - z^k\|_2^2,
\end{aligned} \tag{24}$$

where the first inequality is a consequence of the Lipschitz assumption, the second is a combination of the Lipschitz assumption and the fact that $J$ is skew symmetric. Now using triangle and Jensen's inequalities:

$$\mathcal{H}(z^{k+1}) - \mathcal{H}(z^k) \leq -\epsilon\left(1 + \frac{\mu}{L}\right)\mathcal{H}(z^k) + \epsilon^2 L\left(\|\nabla\mathcal{H}(z^k)\|_2^2 + \|z_\star - z^k\|_2^2\right), \tag{25}$$

where we used the fact that $\|J\nabla\mathcal{H}(z)\|_2^2 = \|\nabla\mathcal{H}(z)\|_2^2$. All together, we have

$$\mathcal{H}(z^{k+1}) - \mathcal{H}(z^k) \leq -\epsilon\left(1 + \frac{\mu}{L}\right)\mathcal{H}(z^k) + 2\epsilon^2 L^2\mathcal{H}(z^k) + 2\epsilon^2\frac{L}{\mu}\mathcal{H}(z^k), \tag{26}$$

Thus, if $2\epsilon \leq (L^2 + L/\mu)^{-1}$, we have

$$\mathcal{H}(z^{k+1}) \leq \left(1 - \epsilon\frac{\mu}{L}\right)\mathcal{H}(z^k) \leq \left(1 - \epsilon\frac{\mu}{L}\right)^k\mathcal{H}(z^0) \tag{27}$$

Taking $2\epsilon = (L^2 + L/\mu)^{-1}$ for simplicity we have

$$\mathcal{H}(z^{k+1}) \leq \left(1 - \frac{\mu}{2L^2\mu + 2L}\frac{\mu}{L}\right)^k\mathcal{H}(z^0) \tag{28}$$

**PDHG corresponds to a discretization of Hamiltonian descent**

The Hamiltonian descent equations are given by

$$\dot{y}_t = \nabla g^*(q_t) - y_t$$
$$\dot{q}_t = -A^T\nabla h(Ay_t) - q_t,$$

and if we parameterize $q_t = A^T p_t$ then we can rewrite these as

$$\dot{y}_t = \nabla g^*(A^T p_t) - y_t$$
$$\dot{p}_t = -\nabla h(Ay_t) - p_t.$$

Now we use the same trick as before, introducing identical terms that we add and subtract

$$\dot{y}_t = \nabla g^*(A^T p_t + y_t/\sigma - y_t/\sigma) - y_t$$
$$\dot{p}_t = -\nabla h(Ay_t + p_t/\rho - p_t/\rho) - p_t,$$

and then discretize as follows (which is valid due to the fact that we assumed that the Hamiltonian was continuously differentiable):

$$(p^{k+\epsilon} - p^k)/\epsilon = -\nabla h(Ay^k + p^{k+\epsilon}/\rho - p^k/\rho) - p^k$$
$$(y^{k+\epsilon} - y^k)/\epsilon = \nabla g^*(A^T p^{k+\epsilon} + y^k/\sigma - y^{k+\epsilon}/\sigma) - y^k.$$

Setting $\epsilon = 1$ and rearranging yields

$$\partial h^*(-p^{k+1}) - p^{k+1}/\rho = Ay^k - p^k/\rho$$
$$\partial g(y^{k+1}) + y^{k+1}/\sigma = A^T p^{k+1} + y^k/\sigma,$$

and finally

$$p^{k+1} = -(I + \rho\partial h^*)^{-1}(\rho Ay^k - p^k)$$
$$y^{k+1} = (I + \sigma\partial g)^{-1}(\sigma A^T p^{k+1} + y^k),$$

which is PDHG.

**Other gradient methods on problem (15)**

(a) HD and proximal gradient descent (PGD).     (b) HD and restarted accelerated gradient (RAG).

(c) HD and conjugate gradient (CG).

Figure 3: Comparison of Hamiltonian descent (HD) and other gradient methods for problem (15) for different $j$.