[Reviews · NeurIPS 2019]

Reviewer 1



To minimize a function f in a continuous time setting, this paper provides an ODE that converges linearly. This ODE is built from the Hamiltonian flow, well known in mechanics, where a perturbation is added. The Hamiltonian is used as a Lyapunov function and the perturbation is necessary, otherwise the Hamiltonian is conserved. However the perturbation is often intractable. Then the authors consider the case where $f$ is composite i.e $f = h \circ A + g$ and build an Hamiltonian from the duality gap of this problem. Using some tricks in the definition of the Hamiltonian, the Hamiltonian dynamics is now tractable. Therefore, the proposed dynamics provide an ODE that exhibits linear convergence of the duality gap to zero. Moreover, a careful discretization of this ODE leads to the well known ADMM. The derivations are elegant and the paper is pleasant to read. The approach of this paper is original and insightful. It provides an new way to look at primal dual algorithms, which are usually seen as discretization of monotone flows, see @article{peypouquet2009evolution, title={Evolution equations for maximal monotone operators: asymptotic analysis in continuous and discrete time}, author={Peypouquet, Juan and Sorin, Sylvain}, journal={arXiv preprint arXiv:0905.1270}, year={2009} } @article{bianchi2017constant, title={A constant step Forward-Backward algorithm involving random maximal monotone operators}, author={Bianchi, Pascal and Hachem, Walid and Salim, Adil}, journal={arXiv preprint arXiv:1702.04144}, year={2017} } or @article{condat2013primal, title={A primal--dual splitting method for convex optimization involving Lipschitzian, proximable and linear composite terms}, author={Condat, Laurent}, journal={Journal of Optimization Theory and Applications}, volume={158}, number={2}, pages={460--479}, year={2013}, publisher={Springer} } ******************************************************************** I appreciate that the authors will cover PDHG algorithm in the camera-ready version. If PDHG can be covered, I expect that Vu-Condat can also be covered. Regarding the references provided in the review, please note that I'm not asking the authors to add all references to their paper (some may be relevant while some other may not). I pointed out these references for future works.

Reviewer 2



Although the paper contains a lot of reminders about the Hamiltonian, it proposes a new research direction that would deserve to be explored. The proposed Halmitonian descent seems indeed to provide better results than the gradient descent. Numerical results in two simple regression scenarios evidence this fact.

Reviewer 3



#### I'd like to thank the authors for their feedback. Since they also managed to obtain primal-dual hybrid gradient using hamiltonian, I increase my score from 4 to 5. For me it's very hard to say whether this work is really a good fit for neurips. To me it's more like pure math, but the other reviewers seems to be very confident that it addressed a very important problem. I'm partially convinced by them, so I put overall score 6 (i.e. an additional increase from 5 to 6 for importance). ### Thank you for your work and for clarity of writing. Unfortunately, I do not see any interesting message in this work. First, let's look at the hamiltonian descent. I agree with the authors that its importance depends on whether we can use it without knowing the optimum and without computing expensive gradient of the conjugate. Ideally, I want to see either a recipe how to do this for different problems or at least a significant number of examples. In this work we only see it for composite optimization, so I do not feel that the approach has generality. The next result, affine invariance, is quite nice as we expect a natural method to be invariant to the choice of basis. However, this property is not important per se and it doesn't seem that the authors use it in any way. There is a brief discussion of discretization. It's simple, clear and it's later used in the discussion of ADMM. Continuous time analysis usually comes with discretization schemes and it's mostly a technical detail. In appendix we see O(1/k) rate of convergence for the explicit scheme, which does not match what we see in the experiments. Then we come to the interesting part, which is composite minimization. Since the duality gap includes explicitly solutions in linear form, they cancel out with the corresponding terms in the hamiltonian descent dynamics. This is nice, because now we can indeed discretize it and obtain a method for composite minimization. This is where we come to ADMM as it turns out it can be obtained as a discretization of hamiltonian descent. To prove it, an extra trick was used, which is to add 0=(x(t) - x(t)) and then discretize it as x^{k+1} - x^k. It's not clear to me if a trick like that is always needed and how the authors came up with it. Probably this is natural to do if one wants to obtain the update of ADMM, but I feel that overall this makes this contribution smaller, because it requires knowledge of what we want to obtain. I don't see any discussion of what happens if we don't use the trick and it wasn't discussed in the section about discretization. And although it is interesting to see ADMM as a discretized version of Hamiltonian descent, we already have an ODE to obtain ADMM and its accelerated variant. There are whole two pages devoted to experiments and I don't think it's so important here. The main contribution is the theory and more theoretical results would be better. Moreover, the experiments do not validate the theory: the explicit scheme in the first experiment numerically converges linearly, while the theory only guarantees O(1/k) rate. The proposed approach outperforms an accelerated method as can be seen in Appendix in Figure 3, which is quite interesting, but there is no explanation for this phenomenon as the rate is not linear. A lot of details provided for experiment 1 could be moved to the appendix to create more space for theory. Second experiments would benefit from comparison to accelerated gradient method or a stochastic method. Let me make a conclusion. The presented theory is quite nice, but it definitely lacks depths. Probably this work is better suited for a mathematical journal, but I do not see how it can interest the NeurIPS community. This does not mean that the authors should give up on their work, there might be other methods that can be discovered using this approach. Alternatively, the authors might try to obtain more existing methods such as Primal-Dual Hybrid Gradient or Condat-Vu as a discretization of Hamiltonian Descent on some other problem. Or, another interesting question is what happens when primal-dual relation breaks because of nonconvexity. In any case, at the current stage the work does not go deep enough to be published.

[Author Response · NeurIPS 2019]

We thank the reviewers for their time, thoughtful reviews, and feedback; the suggestions will help us strengthen the paper. (R1) found the paper to be original and insightful and the derivations to be elegant; (R2) agrees that this is a new research direction that deserves to be explored; (R1) and (R3) noted the clarity of the writing. We thank the reviewers for their kind words. The concerns raised by the reviewers focus on how our methods relate to other methods in the literature and the convergence rates. We address each in turn, along with other minor questions.

**Relationship to other methods.** Firstly, we added *all* suggested references, including related work from Hedy Attouch; we thank the reviewers for bringing these to our attention. We have expanded the discussion of other methods in the literature including Condat-Vu and Hamiltonian perspectives on acceleration. (R3) noted that we might obtain other existing methods as a discretizations of Hamiltonian descent. To address this we have added a derivation of PDHG from the Hamtilonian descent ODE to the appendix; unfortunately Condat-Vu does not fit neatly into our framework (at the moment at least).

(R3) had some concerns about the discretization trick we used to derive a connection between ADMM and the Hamitlonian ODE. One can view it from the other direction—the Hamiltonian ODE can be recovered by taking the step-size in ADMM to zero. The 'trick' we use is standard, see, e.g., the cited work by Wilson, Recht, and Jordan, 2018. Finally, we provide two other 'vanilla' discretizations that do not use that particular trick.

**Convergence questions.** Regarding convergence rates of ADMM, (R1) asked whether this Hamiltonian perspective on ADMM will yield a distinct convergence rate analysis. This is an interesting question that we must leave to future work. Ultimately, each discretization scheme is individually analyzed and ADMM has been extensively studied in the literature where an $O(1/k)$ rate can be obtained, e.g., by He and Yuan, 2012 (we have added this reference too).

(R3) noted that the $O(1/k)$ convergence rate in our analysis is not matched by experiment. Our analysis is a worst-case analysis that does not assume strong convexity - for some problems it may well do better than the worst-case rate. In particular the two examples we presented have (local) strong convexity - we have added an additional proof to the appendix that shows linear convergence under (global) strong convexity. Now the experimental results and the theory match for the strongly convex case.

(R1) suggested extending the analysis to non-smooth Hamiltonians and pointed out some potential difficulties, along with a reference. We agree that this would be very useful work and would like to tackle it in the future, although the difficulties pointed out are significant.

**Additional experiments.** (R2) asked for examples on higher dimensional data. We have added additional runs of the existing experiments scaling to higher dimensions - we include here a sneak preview of the results of the same problem as example 1 where the data is $1000 \times 1000$ instead of $50 \times 50$, in Figure 1.

**Generality of composite optimization.** (R3) was concerned that the composite optimization problem was not sufficiently general to warrant interest. This problem (the sum of two objectives related by a linear mapping) has been studied extensively in the literature and many problems can be written in this form, including cone programs, regularized loss minimization, etc. Since it has so many applications, many algorithms have been developed to solve it, including ADMM, FISTA, PDHG, Condat-Vu etc. We think this is, in fact, one of the most important problems in optimization.

**Importance of affine invariance.** (R3) did not agree that the affine invariance property was important. Affine invariance provides robustness to poor conditioning without explicit knowledge of the conditioning; it is one of the main advantages that second-order methods have over first-order methods. For this reason, it is interesting to have a first-order method (Hamiltonian descent) that is affine invariant. We refer (R3) to Boyd and Vandenberghe, 2004, Sec 9.5 where the importance of this property is discussed. In fact, the message in Figure 1 of the original draft was to demonstrate numerically the practical impact of affine invariance; we generated a sequence of problems with worsening conditioning and showed that both standard and accelerated gradient techniques suffer greatly when the conditioning worsens, but our technique is unaffected. Since our technique is unaffected by the worsening conditioning it would stand to reason that for some problems it would do better than an accelerated gradient technique.

Figure 1: Higher dimensional regularized least squares example. Problem (14) in original draft.

[Meta-Review · NeurIPS 2019]

Congratulations! This paper was a borderline case that led to significant discussion among the reviewers, but in the end we decided the novel ODE-style approach would be of strong interest to the NeurIPS community. In your revision, please address all promises in the rebuttal and comments in the reviews. If there are additional experiments or examples you can include to demonstrate practical value or application of this technique to machine learning tasks, please include these in the revision to better demonstrate where your method might be used---this was the main weakness identified by the reviewers.